# An Insight into Nano Silver Fluoride-Coated Silk Fibroin Bioinspired Membrane Properties for Guided Tissue Regeneration

**DOI:** 10.3390/polym13162659

**Published:** 2021-08-10

**Authors:** Aditi Pandey, Tzu-Sen Yang, Ta-I Yang, Wendimi Fatimata Belem, Nai-Chia Teng, I-Wen Chen, Ching-Shuan Huang, Aivaras Kareiva, Jen-Chang Yang

**Affiliations:** 1Graduate Institute of Nanomedicine and Medical Engineering, College of Biomedical Engineering, Taipei Medical University, Taipei 11052, Taiwan; aditi8293@tmu.edu.tw; 2Graduate Institute of Biomedical Optomechatronics, Taipei Medical University, Taipei 11031, Taiwan; tsyang@tmu.edu.tw; 3Department of Chemical Engineering, Chung-Yuan Christian University, Taoyuan 32023, Taiwan; taiyangyang@gmail.com; 4International Ph.D. Program in Biomedical Engineering, College of Biomedical Engineering, Taipei Medical University, Taipei 11031, Taiwan; belemfatimataw@gmail.com; 5School of Dentistry, College of Oral Medicine, Taipei Medical University, Taipei 11052, Taiwan; tengnaichia@hotmail.com (N.-C.T.); fatima777@tmu.edu.tw (I.-W.C.); jollyhuangtw12@gmail.com (C.-S.H.); 6Institute of Chemistry, Vilnius University, Naugarduko 24, LT-03225 Vilnius, Lithuania; aivaras.kareiva@chgf.vu.lt; 7Research Center of Biomedical Device, Taipei Medical University, Taipei 11052, Taiwan; 8Research Center of Digital Oral Science and Technology, Taipei Medical University, Taipei 11052, Taiwan

**Keywords:** nano silver fluoride, silver diamine fluoride, antibacterial efficacy, biomineralization, in vitro cytotoxicity

## Abstract

The current work focuses on the development of a novel electrospun silk fibroin (SF) nonwoven mat as a GTR membrane with antibacterial, biomineralization and biocompatible properties. The γ-poly glutamic acid (γ-PGA)-capped nano silver fluoride (NSF) and silver diamine fluoride (SDF) were first synthesized, which were dip-coated onto electrospun silk fibroin mats (NSF-SF and SDF-SF). UV-Vis spectroscopy and TEM depicted the formation of silver nanoparticles. NSF-SF and SDF-SF demonstrated antibacterial properties (against *Porphyromonas gingivalis*) with 3.1 and 6.7 folds higher relative to SF, respectively. Post-mineralization in simulated body fluid, the NSF-SF effectively promoted apatite precipitation (Ca/P ~1.67), while the SDF-SF depicted deposition of silver nanoparticles, assessed by SEM-EDS. According to the FTIR-ATR deconvolution analysis, NSF-SF portrayed ~75% estimated hydroxyapatite crystallinity index (CI), whereas pure SF and SDF-SF demonstrated ~60%. The biocompatibility of NSF-SF was ~82% when compared to the control, while SDF-coated samples revealed in vitro cytotoxicity, further needing in vivo studies for a definite conclusion. Furthermore, the NSF-SF revealed the highest tensile strength of 0.32 N/mm and 1.76% elongation at break. Therefore, it is substantiated that the novel bioactive and antibacterial NSF-SF membranes can serve as a potential candidate, shedding light on further in-depth analysis for GTR applications.

## 1. Introduction

Periodontitis, a chronic inflammatory disease, is a condition caused by microorganisms in the tissues supporting teeth, leading to progressive deterioration of ligament, alveolar bone, thereby forming a deep periodontal pocket and gingival recession [1,2]. Periodontium regeneration can be achieved by various clinical therapies opted for treating severe periodontitis lesions. Periodontal defect reconstruction via guided tissue regeneration (GTR) membranes is one of the successful surgical techniques to resist proliferating connective tissue from migrating into the defect site [3].

The component used for the construction of the membrane used for GTR is a vital aspect of its efficacy. Essentially, the membrane should fulfill the main designing criteria for GTR, such as biocompatibility, cell occlusion, spaciousness, clinical manageability, and tissue integration [4]. Regarding the membrane materials used, various examples include synthetic polymers such as polytetrafluoroethylene (PTFE), polylactic acid (PLA), polyglutamic acid (PGA), natural polymers such as collagen, chitosan, gelatin, etc. [5,6]. The resorbable materials offer the advantages over the non-resorbable membranes (such as PTFE, titanium mesh, etc. [3]) of no further surgical interventions to remove the membrane, and therefore are extensively used in different clinical situations [6]. A popularly used natural polymer, silk fibroin (SF; drug-loaded, alone or functionalized) has noteworthy mechanical properties, well-recognized biocompatibility, proteolytic degradability, minimum inflammatory reaction, and is cost-effective. Moreover, the efficacy of silk fibroin towards protein and cell response mimics the extracellular matrix collagen, through the production of nanofibers by the electrospinning process [7,8,9]. However, the disadvantage of the produced SF lies in its brittleness, which restricts its utilization for flexible membrane material [10,11]. Therefore, SF was blended or reinforced with other materials such as polyethylene oxide and chitosan [12] to achieve desired mechanical or biological properties. However, the intended material of use for this work, SF, lacks antibacterial property for biomedical applications, for which, SF was enriched/modified by the use of functional nanomaterials/carriers such as silver, chitosan, graphene, and graphene oxide [13,14,15,16,17].

The key to successful therapy involves the reduction/elimination of periodontal pathogens (anaerobic bacteria) from the deep periodontal pocket [1,18]. Bacteria such as *Aggregatibacter actinomycetemcomitans* and *Porphyromonas gingivalis (*P.gingivalis*)*, occur in the sub-gingival plaque/biofilm [19]. Microbial contamination of the GTR surgical sites may impact the cellular attachment process for periodontal ligament [20,21] and jeopardize the outcome of treatment, in terms of a hindrance towards the connective tissue/bone formation [22]. It was observed that introduction of antibiotics such as amoxicillin or tetracycline onto the membranes greatly reduced the bacterial adhesion [23]. Silver nanoparticles (AgNPs), developed by various techniques, result in higher anti-microbial activity in treating infections (due to unique chemistry, shape, and higher surface area) [24,25]. In tackling the infection, their applications include endodontics, dental, bone implantology and prostheses, and GTR [24,26,27,28,29]. The effect of AgNP-GTR has even been compared with the doxycycline (DOX-GTR) and found to be effective against several bacteria associated [29]. Silver diamine fluoride (SDF) and nano silver fluoride (NSF) are amongst the effective antibacterial agents used for treating dental caries [30,31]. Furthermore, fluoride also promotes remineralization. The efficacy of nano-silver constituting the fluoride inclusion was considered equivalent to that of SDF, with an added advantage of lower cytotoxicity than the SDF [30]. It is also suggested that the efficacy of NSF may be better than conventional fluorides regarding the treatment of caries lesion owing to the antibacterial action and remineralization [32]. If indicated, there may be a need for a material acting as a source of calcium or enhancing mineralization and aiding particle retention in the early process of GTR, which may also help engineer new periodontium, while restricting the gingival growth. The fabrication of such a kind of functional periodontal membrane by the electrospinning technique may act as an implant/interface between tissue and bone [33,34,35].

To date, no work has been reported for the application of NSF/SDF-coated materials in the field of guided tissue regeneration. Therefore, considering these facts, this work aims for the development of an NSF-coated electrospun SF and compared it with SDF-SF. The coated-SF materials are believed to act as an efficient antibacterial material. Moreover, this material may supposedly enhance tissue regeneration under the biomineralization process and possess efficient mechanical properties.

## 2. Materials and Methods

### 2.1. Materials

Starting powders, silver nitrate (99.85); sodium borohydride (98+%); ammonium fluoride (96%); ammonia water (25%); sodium carbonate (99%); anhydrous lithium bromide (99%); ammonium persulfate (98%); aniline monomer (99%); formic acid (98%); and *Bombyx mori* silkworm cocoon were used as received without further purification. All reagents were obtained from Sigma-Aldrich, New Taipei City, Taiwan.

### 2.2. Preparation of Degummed Silk Fibroin

Native silkworm (*Bombyx mori*) silk comprises dual silk fibroin fibers coated with sericin. The sericin was removed by a degumming process, which implies boiling the silkworm cocoons in 0.02 M sodium carbonate solution, for 30 min. Subsequently, the dried degummed silk fibers were dissolved in a 9.3 M lithium bromide solution for 4 h at 60 °C. The resultant fibroin solution was dialyzed using a dialysis cassette with 3500-MW cutoff (MWCO) for 72 h, and then lyophilized for long-term storage.

### 2.3. Preparation of Silk Fibroin Nanofiber Mats

Silk fibroin nanofiber mats were prepared by using an electrospinning apparatus detail reported elsewhere [36]. The formic acid was utilized to prepare 10 wt% silk fibroin solutions. Subsequently, the resulting solution was ejected through a syringe with a needle size of gauge 20. The feeding rate, applied voltage, and tip-target distance for electrospinning were 0.15 mL/hr, 18 kV, and 12 cm, respectively. The fabricated electrospun SF mats were treated with 90% ethanol, overnight, and then air-dried.

### 2.4. Synthesis of NSF and SDF, and Their Coating on SF

The AgNPs were synthesized by the reduction reaction of silver nitrate (AgNO_3_) using sodium borohydride (NaBH_4_) and γ-PGA as a capping agent [30]. Typically, we homogenized AgNO_3_ (1 mL, 0.11 M) and γ-PGA (27.5 mL, 1.7 mM), initially dissolved in de-ionized water (DI) water, under magnetic stirring. Then, the mixture was subjected to drop-wise addition of freshly prepared NaBH_4_ (0.5 mL, 0.3 M), while stirring vigorously. The Ag^+^ ion reduction was triggered, and there was a change in the color of solution from colorless to reddish brown. Ammonium fluoride (NH_4_F) (1 mL, 0.21 M) was added, under constant stirring, overnight. The synthesized solution was 1% NSF (represented as NSF throughout the manuscript), a dark brown colored solution, which was stored at 4 °C in the refrigerator in a bottle covered with black plastic. The UV-Visible spectroscopy (JASCO V-770 Spectrophotometer, Tokyo, Japan) was then performed for the confirmation of the formation of AgNPs, and the size was estimated by transmission electron microscopy (TEM HT7700, HITACHI, Tokyo, Japan).

The 38% SDF was prepared by adding 6.8 g of AgNO_3_ in 5 mL of DI water, mixed homogenously by vortexing. This was followed by NH_4_F addition (1.47 g), and again homogenized. Then, ammonia was introduced to the solution and the final volume of 13.2 mL was reached with DI water, keeping pH 7. The colorless solution was stored in a bottle (wrapped with a black plastic) at 8 °C [37]. The 38% SDF (concentrated SDF, represented as CSDF) was also diluted to 1% SDF (diluted SDF, represented as DSDF), in order to have the same concentration as that of NSF (for comparison purposes).

The prepared NSF and SDF (CSDF, DSDF) were coated using the dip-coating process on the SF mats for a period of 24 h under dark conditions and then oven-dried and stored in the dark.

### 2.5. Antibacterial Activity of the Coated SF

The antibacterial property of the samples (SF, NSF, CSDF, DSDF-coated silk fibroin, *n* = 3) was assessed using Gram-negative (*P.gingivalis*; ATCC 33277). In a typical experiment, the samples were introduced into a 24-well plate (in two sets), and 100 μL of the bacterial solution (0.1 optical density, OD) was seeded upon samples. The samples were incubated under anaerobic conditions, for 48 h, at 37 °C, after which, they were thoroughly rinsed with PBS to remove the non-adherent bacterial cells. One of the sets of the cultured samples was analyzed by SEM imaging for qualitative and morphological analysis. The adhered bacteria were fixed with glutaraldehyde and incubated at 4 °C overnight. After thorough washing with PBS, the samples were then dehydrated in a gradient alcohol series and sputter-coated with gold for scanning electron microscopy (SEM, SU3500, HITACHI, Tokyo, Japan) analysis. On the other set of cultured samples, the MTT analysis was carried out and the absorbance was measured at 570 nm, for the quantitative assessment.

### 2.6. Biomineralization Studies

Simulated body fluid (SBF) was utilized as the immersion solution to perform the biomineralization process on the samples. The 10× SBF was prepared by Tas and Bhaduri method [38]. The chemicals and their sequence of mixing were according to Table 1, starting from NaCl and ending with Na_2_HPO_4_. Just before the biomineralization experiment was conducted, NaHCO_3_ was added to the solution and homogenized under constant stirring. The 10× SBF was stored at 4 °C.

The electrospun samples (SF, NSF, CSDF, DSDF-coated SF, *n* = 3) were cut into 5 × 5 mm pieces, placed in 24-well plates and soaked into the 10× SBF for 24 h under continuous stirring. After this, they were oven-dried, gold-coated and subjected to microstructural and elemental characterization under SEM-EDS.

The chemical characterization of the formed apatite crystals on the samples was performed by the FTIR-ATR spectroscopy (Nicolet 6700, Thermo-Fisher-Nicolet Instruments, Madison, WI, USA, coupled by mercury-cadmium-telluride (MCT) detector and infrared synchrotron radiation as light source at BL14A1, National Synchrotron Radiation Research Center (NSRRC) Taiwan, for 128 scans, resolution of 4 cm^−1^, and compared with the FTIR spectra of the non-biomineralized original silk, and coated-silk samples. The spectra were deconvoluted and peak fitting was carried out using peak fit software (Peakfit v. 4.11, Systat Software, Inc., San Jose, CA, USA).

### 2.7. Biocompatibility Studies of the Extracted Medium and SF-Based Samples

The mouse fibroblasts cell line, L929 cell line (BCRC, RM60091, Bioresource Collection and Research Center, Hsinchu City, Taiwan). The cells were first expanded in Dulbecco’s modified eagles medium with 10% FBS (Gibco 10082147, ThermoFisher, Tokyo, Japan) and 3 μg/mL geneticin (G418, ThermoFisher, Tokyo, Japan) at 37 °C in a 95% humidity and 5% CO_2_ incubator.

In a typical experiment, following a modified protocol of ISO10993-5 method [39], 1 × 10^6^ cells/mL cells were incubated with extract of specimens (*n* = 3, 5 × 5 mm, earlier incubated for 72 h at 37 °C for extraction process), in 24 well culture plate and allowed to incubate for 24 h. Then, the medium was removed, and the wells were rinsed with phosphate buffer saline (PBS). In order to determine the cell viability (through the metabolic activity determination), MTT assay (MTT: (3(4, 5-dimethylthiazol-2-yl)-2,5-diphenyl tetrazolium bromide)) (Sigma Aldrich, St. Louis, MO, USA), was performed, for the quantitative estimation. MTT: PBS (1:10 ratio) was then added to each well, and incubated for 4 h, after which dimethyl sulphoxide (DSMO) was added to the wells, thereby dissolving the formazan crystals to develop a purple color. The absorbance of the formed, purple-colored solution was read at 570 nm, by a microplate reader.

### 2.8. Mechanical Property Measurements

Tensile tests of the electrospun SF-based samples (dimension of 30 × 10 mm) were carried out with a TA.XT plus Texture Analyser, Texture Technologies, Hamilton, MA, USA, with a 50-N load cell, at a strain rate of 10 mm/min. The samples were mounted vertically between the gripping units of the mechanical testing apparatus, with a gauge length of 20 mm for mechanical loading. Due to the porous nature of nonwoven, we were not able to have the exact area of the cross-section to have the value of force/area (F/A). Therefore, instead of F/A, we can use the F/width (N/10 mm) [40].

### 2.9. Statistical Analysis

The experiments were in triplicates, with mean ± standard deviation calculation, and statistical significance determination by Student’s *T*-test (*p* < 0.05).

## 3. Results and Discussions

### 3.1. Characterization of the Silver Nanoparticles

The UV-Vis absorption spectra of the AgNPs (NSF) is represented in Figure 1a. The characteristic peak at 403 nm depicts the formation of AgNPs from AgNO_3_ (in the presence of γ-PGA) upon addition of NaBH_4_, indicating reduction of Ag^+^ ions to Ag^0^, which shows the characterization of the synthesized AgNPs. The TEM micrographs in Figure 1b reveals nano-sized spherical particles, ranging in between 3–6 nm, and capped by γ-PGA.

The current work involved the development of polyelectrolyte-based (involving electrostatic interactions with positive silver ions) formulation, comprising of nano silver fluoride, NSF. As reported, chitosan as a carrier was used for the NSF synthesis because of its ability to form complex interactions with AgNPs [30] and proteins [41], thereby increasing muco-adhesion. However, the γ-PGA used as an anionic polyelectrolyte is assumed to interact electrostatically with the cationic Ag ions, thereby forming a stable complex.

### 3.2. Microstructural and Chemical Characterization of Uncoated and Coated SF

The colloidal AgNPs (NSF) and the elemental silver in SDF, after preparation, were subsequently coated on electrospun SF webs, by overnight coating and then drying. The AgNPs on SF were detected and confirmed by their morphological investigations through SEM. Figure 2 represents SEM images of uncoated SF, and elemental Ag and AgNPs coated SF (indicated by yellow arrows). The nonwoven electrospun nanofibres were reported for preventing the cell infiltration, leading to occlusion, without depleting oxygen and nutrients, thus justifying the role of electrospinning as a promising method for developing physical barrier. The NSF/SDF coated SF shows uniform distribution on the coated electrospun fiber mats [42]. It is suggested that the high surface free energy of silver nanoparticles, which leads to their instability, causes migration and aggregation of silver nanoparticles [43].

### 3.3. Antibacterial Efficacy Evaluation of SF-Based Samples

A Gram-negative and pathogenic bacteria *P.gingivalis*, occurring in the sub-gingival plaque/biofilm, was utilized to examine the antibacterial activity of silk-based membranes. The intertwined membranes after their immersion into bacterial suspension for 24 h, gave rise to *P.gingivalis* adhesion and growth onto the surface, as presented in the SEM images in Figure 3a–e. There is no adhesion of bacteria on the pure SF only when immersed in the medium without bacteria (only medium and SF, MS), portraying no microbial contamination of the medium. Further, upon exposing the uncoated SF to *P.gingivalis* broth, the membrane surface (medium with silk fibroin and bacteria, MSB) is densely covered by bacterial cells. On the other hand, the CSDF-SF sample shows negligible bacterial adhesion. However, when diluted or 1% SDF was exposed to the bacteria, it can be observed that the lowered silver content in DSDF samples apparently may not be enough to act against the bacteria, leading to increased bacterial adhesion than that on CSDF-SF, however lower than that on the pure SF sample. The NSF-SF sample displays effectively lower bacterial density when compared to SF and DSDF matrices, although not as much as the CSDF-coated matrix.

The metabolic activity (hence the bacterial viability) of the *P.gingivalis* bacteria on the electrospun silk fibroin-based samples (wherein the coated SF samples are denoted as CSDF, DSDF, and NSF) as assessed by the MTT assay (Figure 3f), is found to be in the same trend corresponding to the SEM images. The highest bacterial viability is on the SF sample immersed in bacterial broth, which is considered to be 100%. From these findings it can be noted that the 38% SDF coated sample (CSDF) is found to inhibit maximum bacterial growth (6.7 folds lower than SF), attributing to its highest silver content. However, the NSF-SF sample is also found to be a potential material controlling the *P.gingivalis* adhesion, as portrayed by the significantly lower bacterial density of 3.1 and 1.9 folds, when compared to SF and DSDF (1.6 folds lower than SF), respectively, yet 2.1 folds significantly higher density relative to CSDF.

Apart from the role of SDF as a popular anti-caries agent [44], as recently reported by Rams et al., 38% and 19% of SDF considerably inhibited in vitro activity of sub-gingival pathogens extracted from severe periodontitis lesions, suggesting its potential in periodontal infection management [45]. The silver ions have a broad spectrum of antimicrobial activity. Their mode of action involves attacking the -SH groups of enzymes, obstructing the pathway of protein synthesis. These ions also denature the bacterial DNA, causing a bactericidal action, and were used as reinforcements/coatings [46,47,48]. Another widely used caries inhibiting formulation is NSF with antimicrobial activity similar to SDF, which has the advantage of its lower toxicity to eukaryotic cells on account of utilizing silver nanoparticles [30]. The AgNPs, when synthesized using an appropriate capping agent may be effective in controlling periodontal infections, with the added advantage of smaller size as a criterion for enhanced antibacterial action against oral anaerobic pathogens [49]. The GTR dressing coated with AgNPs leads to a reduction in bacterial adherence or penetration, which may aid in the treatment of intra-bony defects and clinical success [50,51].

### 3.4. Effects of SBF on Coated and Uncoated Silk Observed by SEM-EDS and ATR-Spectroscopy

The apatite deposition on the silk fibroin-based membrane was proceeded by immersing them into 10× SBF solution (for 24 h) for the biomineralization process as shown by the SEM images (Figure 4a–d) and its corresponding elemental analysis spectra in (Figure 4e–h). The molar Ca/P ratio of a porous and entangled matrix of pure SF (~2.1), CSDF-coated (<0.5, 80%Ag), DSDF-coated (<0.5, 7.3% Ag) and NSF-coated SF (~1.67, negligible Ag) following 24 h SBF immersion depicts high Ca and P deposition on pure SF and NSF-SF samples, while negligible Ca, P and high Ag deposition on SDF-coated samples. Silver is less reactive than hydrogen and can hardly replace H^+^ in a reaction. Thus Ag^+^ is inclined to bind to a specific site of MMPs to inactivate their catalytic functions. The silver ions from SDF are also supposed to inhibit the metal metalloproteinase (MMP) and cathepsin, thereby preventing collagen degradation [52].

The Ca/P ratio obtained for the SF sample, higher than the ideal molar ratio of Ca/P of hydroxyapatite (1.67), and that for the NSF-SF membrane can also be seen by Ca, P, and O distribution in the elemental mapping images shown in Figure 5. The high Ca/P ratio demonstrated calcium phosphate layer formation, resembling the mineral content of natural bone and verified the biomimetic apatite formation [53]. The SDF-treated samples present Ag deposition on the samples after biomineralization. The concentrated (38%) and diluted (~1%) SDF-SF samples result in ~80% and ~7% precipitation of silver. The aggregation of Ag is in the same line with in vitro studies reported for the reaction of SDF with gelatin (as a collagen substitute) for deposition of metallic silver [54,55]. The hydroxyapatite crystal deposition on the γ-PGA NSF-SF possesses a molar Ca/P ratio of ~1.67. The high apatite deposition on this sample may be accredited to the effect of γ-PGA-capped NSF on the silk fibroin. The glutamic acid residues possess a negative charge, which attracts more Ca^+^ ions from SBF, thereby attracting the PO_4_^−^ ions and enhance the supersaturation such that they develop into HAp crystals [56,57].

Further, in order to analyze the chemical composition of the samples, FTIR-ATR spectroscopy was conducted. As shown in Figure 6, the ATR spectra of SF-based samples depict the apatite deposition in the form of characteristic HAp absorption peaks/bands observed at ~1130 cm^−1^ specifically corresponding to the P–O stretching of the PO_4_ [58,59]. This peak is observed for all samples, SF, CSDF-SF, DSDF-SF, and NSF-SF. Furthermore, the additional distinctive peaks of PO_4_ are found to be appearing from 950–960 cm^−1^. The apatitic phase at ~960 cm^−1^ relates to the γ_1_ (PO_4_) of the HAp, which may be of lower intensity. Indeed, the 950 cm^−1^ peak portrays the amorphous calcium phosphate (ACP) formation [59].

The estimated HAp crystallinity index (CI) is estimated as the ratio of absorbance A_960_ to (A_960_ + A_950_), which is found to be a maximum of ~75% for NSF-SF samples, depicting the highest crystalline apatite deposition corresponding to the SEM-EDS data. Further, the SF membranes elicit ~60% estimated HAp CI, and as also observed through the SEM-EDS, the Ca/P ratio was ~2.1, because of which there may be a possibility of the presence of ACP phase, which may not have completely transformed to HAp mineralization process, giving rise to the high Ca/P ratio of ACP and HAp together. Noteworthy, in the case of CSDF-SF samples, there occurs ~60% estimated HAp CI, which was not observed from the SEM-EDS. This difference in the inference may be possibly attributed to the masking effect of the silver ions (~80%) on the sample surface after the mineralization reactions. The effective biomimetic mineralization invoked post 10× SBF treatment suggests the possible bone-regeneration platform (owing to bone-bonding potential and biocompatibility) offered by these electrospun SF-based constructs, with the highest estimated CI on NSF-SF [60,61,62].

The β-sheet content is assessed by the ratio of absorbance of β-sheet (1265 cm^−1^) to the sum total of absorbance of β-sheet (1265 cm^−1^) and of the random coil (1235 cm^−1^), as represented in Table 2, along with their full-width half maxima (FWHM). Appendix A represents the FTIR-ATR spectra of the SF-based samples before biomineralization. Appendix A shows the β-sheet/random coil ratio and their corresponding FWHM values. An enhancement by ~1.7 folds of the β-sheet proportion before and after mineralization for SF, and NSF-SF samples is revealed, while for the SDF-coated samples, there is barely a difference noted.

### 3.5. Response of Extracted Medium from the SF-Based Samples towards L929 Cell Line

The cytotoxic properties of the membrane-extracted substrates are represented in Figure 7. It can be demonstrated that extract of SF (90.15 ± 7.02%) and NSF-SF (78.06 ± 4.00%) are not cytotoxic in comparison to the control group (comprising cells and medium only). On the contrary, the DSDF and CSDF-SF extracts elicit only 8.97 ± 1.36% and 11.61 ± 3.40% viable cells, respectively, thereby displaying a cytotoxic phenomenon.

An effective understanding of material-host interactions in terms of their biocompatibility is required for the functionality of a biomedical membrane. The excellent biocompatibility of natural silk-based materials has led to its increasing interest in medical purposes. Further, the silk fibroin (SF) gels, micro- and nanonets, sponges, films have the potential ability in therapeutic applications [63,64,65]. Therefore, the cellular response to a newly-developed NSF/SDF-coated SF barrier membrane was investigated using the L929 cell line. The cytotoxic effects of the extracts [66] from both the SDF-coated membranes could be justified by the role of Ag ions released during the 3-day pre-incubation of the membranes with the medium. The role of 0.01% SDF on human gingival fibroblasts triggered almost 100% cell death, depicting its potential cytotoxicity [67]. The SDF used in this study is 38% (CSDF) and 1% (DSDF), which has a higher concentration of Ag, thereby releasing more silver ions into the medium after exposure of 3 days. This accounts for the SDF-induced cell death. However, the extracts from NSF-SF support the cell growth (almost like SF and control samples), despite the presence of nano silver. This could be plausibly attributed to the low concentration of Ag ions, which may have been released after the 72 h exposure time. Therefore, the DSDF and SDF-SF samples raise concerns regarding their in vitro cytotoxic properties. However, the long-term effects of the use of SDF in vivo are still required for further conclusive statements [67]. SDF has also been commonly used in the field of dentistry, which accounts for its no potential risks in humans, due to the combined action of several factors as mentioned before. According to this study, the NSF-SF membranes can be concluded to possess no deleterious effects or cytotoxicity against L929 and the biocompatibility of SF remains unchanged by the NSF-coating on it.

### 3.6. Mechanical Property Analysis of the SF-Based Samples

The tensile strength of the coated and uncoated SF samples is represented in the form of stress–strain curve in Figure 8. It can be observed that the tensile strength of SF samples is 0.22 N/mm, which increases upon addition of NSF by 1.47 folds. Meanwhile, the coating of DSDF and CSDF onto SF reduces the tensile strength to 0.09 and 0.02 N/mm, respectively. Similarly, the elongation at break, indicating the elasticity of the membrane is found to be highest for the NSF-SF (1.76%), followed by SF (1.51%), DSDF-SF (0.67%), and CSDF-SF (0.99%).

Mechanical properties are an important criterion for determining the applications of materials. The variation in mechanical properties of materials should be compatible with the healing or regeneration process. The silk fibroins are characterized as natural polymers, with β-sheets or crystals responsible for stability and in turn the mechanical properties [68]. In electrospun silk fibroins (with higher mechanical properties), treated with ethanol, the β-sheet structure can be easily enhanced [68]. So is observed in our study, wherein the ethanol-treated SF which when further coated by various materials exhibited higher β-sheet content and crystallinity. The enhanced tensile strength of NSF-SF samples could be attributed to the higher crystallinity than the SF samples. This may also be accredited to the strong protective effect of low concentration of silver nanoparticles [69] (in NSF), distributed evenly throughout the SF matrix. This may help in the possible closure of the certain voids present in the membrane by the NSF particles, thus leading to the strong reinforcing effect by the AgNPs, inhibiting polymer drawing, and increasing the tensile strength. However, the lowered tensile strength of the CSDF and DSDF-SF samples, despite the higher crystallinity may be due to the high concentration of the silver ions in the DSDF and CSDF solutions, which likely led to the agglomeration and aggregation of Ag^+^ ions onto the SF membranes. This, in turn, may lead to the stress concentration effect, thereby deteriorating the tensile strength of the samples [69]. Therefore, it can be concluded that the NSF-SF samples can be considered with the suitable balance between the mechanical properties, which may behave as an apt candidate for GTR applications, after in-depth analysis.

Scheme 1 represents the space making NSF-SF GTR membrane (and its required properties), placed at the affected area, (a) its fabrication (from electrospinning of silk fibroin to NSF dip coating on the SF membrane), subjected to in vitro studies, (b) antibacterial efficacy (against *P.gingivalis*), (c) bioactivity (mineralization in 10× SBF) analysis, (d) biocompatibility analysis (cytotoxicity study by reaction of MTT reagent with L929 cell), and (e) mechanical strength test (tensile stress and elongation at break). In this work, the effective antibacterial activity of NSF, comparable with SDF, against *P.gingivalis* speculates its budding potential in sub-gingival applications.

## 4. Conclusions

The NSF/SDF-coated electrospun silk fibroin membranes were assessed for their biological properties in GTR applications. The biomineralization process of the SF-based samples demonstrated apatite precipitation. The antibacterial efficacy against *P.gingivalis* concluded the bactericidal nature of all the silver-containing compounds when compared to silk fibroin alone, with the NSF/SDF-coated samples as the most efficient amongst all samples. The SEM-EDS analysis depicted Ca/P precipitation on NSF-SF and SF samples. While the detailed surface analysis by FTIR-ATR spectroscopy portrayed the hydroxyapatite deposition on all the samples, as seen by its corresponding peaks. The NSF-SF membranes depicted the highest estimated CI, suggesting excellent bioactivity. Moreover, the tensile strength of the NSF-SF samples was found to be the highest. Therefore, this study corroborates the potent bioactive and antibacterial action of the novel NSF-coated silk fibroin materials as GTR membranes.

## Data Availability

Not applicable.

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
