# Peer review of "An Insight into Nano Silver Fluoride-Coated Silk Fibroin Bioinspired Membrane Properties for Guided Tissue Regeneration"

_polymers, 2021, doi:10.3390/polym13162659_

Round 1

Reviewer 1 Report

On request of Polymers, I have revised the manuscript titled “An Insight into Nano Silver Fluoride-coated Silk Fibroin Bioinspired Membrane Properties for Guided Tissue Regeneration” by Aditi Pandey and colleagues.

The issue afforded by the authors in this study is the periodontal disease, which corresponds to the deterioration of tooth-supporting tissues, caused by accumulation of bacterial biofilms, and which needs therapies based on the use of guided tissue regeneration (GTR) barriers (membranes) in averting the ingrowth of soft tissues into defect sites and space-making for bone regeneration. Obviously, both inhibition of bacterial growth and biomineralization are necessary to help the tissue regeneration process. Therefore, the authors have thought to improve properties of pristine silk fibroin (SF), a polymer extensively used to develop such barriers, but affected by some drawbacks such as brittleness, with lab-made Ag and fluoride-containing nanoparticles (NPs) designed to possess antibacterial, biomineralization and biocompatible properties, as well as good mechanical characteristics. To this end, the authors prepared AgNPs (containing elemental Ag0), caped them with poly glutamic acid (PGA), which allowed the complexation with fluoride added as NH4F, obtaining NSF NPs, and used them to coat electrospun silk fibroin mats, obtaining NSF-SF. Parallelly, they prepared silver diamine fluoride (SDF) powders by using AgNO3 (containing Ag+) and NH4F and coated SF, as well, achieving SDF-SF. SF, NSF-SF and SDF-SF have been characterized and tested for the antibacterial properties against P. gingivalis, their post-mineralization effects, biocompatibility and mechanical properties obtaining results that asserted the higher performances of NSF-SF in respect of SF alone or SDF-SF.

Considering the relevance of the periodontal disease, which negatively affect the quality of life of humans, the efforts of authors in studying strategies to limit its development and to promote the individuals recover when the disease is already in act, is appreciable. The topic of this work is interesting and further research in this direction is urgent. Also, the results concerning the properties of the materials proposed by the authors are appealing, mainly if compared with those of pristine SF or not nanotechnologically modified agents SDF-FS.

I am for a future publication of this study, and I will reconsider it but, only after that the authors will have addressed the several minor and major issues that affect this original version.

The abstract is too long (close too 300 words), thus not respecting the instruction for the authors of Polymers which require max 200 words. Please, shorten it accordingly.

Line 37. At the first mention, the names of bacteria must be completely written.

The keywords, not in capital letters.

The introduction needs improvements. It would be nice inserting two Tables. Table 1, collecting the criteria requested for membranes applied for periodontal disease treatment, the most used polymers. For each one, advantages and disadvantages, properties, and related references, should be inserted. Table 2, collecting the most important information regarding SF.

In materials and methods section, the authors should insert detailed experimental procedures adopted to perform FTIR and UV-Vis analyses.

Please, add details concerning the physical aspect/state of all prepared material and the storage locations and conditions.

Lines 132 and 143. Please, specify ID water.

It would be appreciable if the authors evaluate the particle size by performing the DLS analysis which allow to obtain also information concerning particles DPI and zeta potential.

To evaluate the Ag0 content by ICP analysis should be performed.

Have authors considered to prepare AgNPs in a “greener” way, by using plant extracts? Why did they select the reductive procedure using NaBH4?

I apologise in advance to the authors but, I did not find the procedure used to coat SF in 2.4 section. In my opinion it is missing.

Line 153. Please define OD.

Line 166. Please, correct “is” with “were” and the “I” of Table in “1”, paying attention to update the Table numbering according to my initial requests.

Line 233. Please remove “as-“, and apply the same along all manuscript.

Section 3.2. In my opinion the authors have not clear what the “elemental silver” is. Differently from what they have asserted, the elemental silver is the silver in its metallic state, which is an insoluble solid where Ag is as Ag0. In SDF there is not elemental Ag but the Ag+ ion. Please correct where necessary.  

Line 253. Please, correct “gram negative” in “Gram-negative”.

Line 261. Please, add “by” at the beginning of the row.

Concerning the authors discussion about the FTIR-ATR analysis of the materials of this study before and after biomineralizations, the authors could obtain more reliable useful information by processing chemometrically the matrix consisting of the spectral data of all the samples. The principal components analysis (PCA) is a useful tool widely applied to interpret the FTIR results. The authors should include PCA in the herein study.

The Section 4 heading not in italic, but bold.

All the reference list does not respect the template of Polymers and must be corrected accordingly.

 On these considerations, I have asked for major revision.

Reviewer 2 Report

  1. Authors need to add more literature in the introduction section to highlight the research gap.
  2. In section 2.1, the authors should add the physical and chemical properties of all chemicals.
  3. In section 2.4, the authors need to visualize the synthesis of NSF and SDF and their coating for better understanding.
  4. In figure 4, the EDS spectra are not clear, improve the quality of figures.
  5. In figure 5, it is hard to read the scales on the figures.

Round 2

Reviewer 1 Report

Althought the authors have not addressed some relevant my requests, the work has been improved and I approve the publication.